# FROGS: a daily 1°x1° gridded precipitation database of rain gauge, satellite and reanalysis products

5  Rémy Roca[1], Lisa V. Alexander[2,3], Gerald Potter[4], Margot Bador[2,3], Rômulo Jucá[5], Steefan Contractor[2,6]

Michael G. Bosilovich[4] and Sophie Cloché[7]

[1]Laboratoire d'Etudes Géophysiques et d'Océanographie Spatiales, Toulouse, France

[2]Climate Change Research Centre, UNSW Sydney, Australia

10  [3]ARC Centre of Excellence for Climate Extremes, UNSW Sydney, Australia

[4]NASA Goddard Space Flight Center, Greenbelt, Maryland

[5]Geoscience Environnement Toulouse, Toulouse, France

[6]School of Mathematics and Statistics, UNSW Sydney, Australia

[7]IPSL, Palaiseau, France

*Correspondence to*: Rémy Roca (remy.roca@legos.obs-mip.fr)

Submitted to ESSD

Abstract.

We introduce the Frequent Rainfall Observations on GridS (FROGS) database (Roca et al. 2019). It is composed of gridded daily precipitation products on a common 1°x1° grid to ease intercomparison and assessment exercises. The database includes satellite, ground–based and reanalysis products. As most

5   of the satellite products rely on rain gauges for calibration, unadjusted versions of satellite products are also provided where available. Each product is provided over its length of record and up to 2017 if available. Quasi-global, quasi-global land only, ocean only as well as tropical only and regional products (over continental Africa and South America) are included. All products are provided on a common netCDF format that is compliant with CF and ACDD standards. Preliminary investigations of

10   this large ensemble indicate that while many features appear robust across the products, the characterization of precipitation extremes exhibit a large spread calling for careful selection of the products used for scientific applications. All datasets are freely available via an ftp server and identified thanks to the DOI: 10.14768/06337394-73A9-407C-9997-0E380DAC5598.

# 1. Introduction

Precipitation is a key element of the water and energy cycle. Observational efforts to document precipitation have a long history (Park et al., 2017) and have matured rapidly in recent decades. Historically available *in situ* archives and associated gridded products (e.g., Becker et al. 2013) are being complemented by the burgeoning capability of satellite observations (Levizzani et al., 2018). Reanalyses precipitation is not only derived from the model physics, but from a short term forecast, and reflect observed atmospheric variability (Bosilovich et al., 2008). While monthly and/or large spatial scale observations have been available for some time, recent progress permits documentation of precipitation at finer space and time scales that consequently allow us to address new challenges such as extreme precipitation (Westra et al., 2013).

Indeed, recent processing of global scale ground-based archives has led to new datasets at the 1°×1° spatial resolution and daily frequency (Contractor et al., 2019; Ziese et al., 2018). A large variety of satellite-based estimates of precipitation at various scales and over various record lengths and spatial coverage have been or are being produced by a growing number of agencies, laboratories and consortiums as monitored and summarized within the International Precipitation Working Group (IPWG) (Levizzani et al., 2018). Finally, the reanalysis community is sustaining a large set of reanalyses, offering products at the 1°×1° daily scale (Potter et al., 2018). This large, unique collection of gridded observational datasets opens the possibility of building an ensemble of products similar to the approach taken by the coupled climate model community. This avoids the need to choose only one product for research applications such as model evaluation. However, it is also important to assess the robustness of the various data sources and understand their differences in order to appreciate the uncertainties within the ensemble and guide its formulation.

Recent GEWEX led assessments have paved the way for efficient and useful coordinated inter comparison/validation exercises: the Cloud Assessment (Stubenrauch et al., 2013) and the Water Vapor Assessment (Schröder et al., 2016, 2019). One of reasons for these successes lies in the making of a common gridded database, facilitating the handling of many datasets that were originally available on various grids, resolutions, formats, data types, etc. The Cloud Assessment database encompasses 23 products, mainly remotely sensed, and is now being extended in time and in the number of products available (Stubenrauch, 2018 personal communication). Within the Water Vapor Assessment, precipitable water spans 22 products, originating from *in situ* observations, satellite measurements and reanalysis output (Schröder et al., 2018). In both assessments though, a common publicly available database has been provided for further analysis after the main assessment effort. Here we follow the legacy of the GEWEX Data Analysis Panel (GDAP) assessments by building a database of daily precipitation with data originating from rain gauges, satellite and reanalysis products. The database, Frequent Rainfall Observations on GridS (FROGS), is released at the beginning of the assessment. This will help to integrate new investigations and its overall assessment recently initiated under the auspices of GEWEX/GDAP and IPWG (Haddad and Roca, 2017). This also includes a dedicated effort to analyse extreme events and assess their characteristics under the joint WCRP Grand Challenge on Weather and Climate Extremes/GEWEX GDAP project on "Precipitation Extremes" (Alexander et al., 2018), with a Special Issue being developed on the topic.

The aim of this paper is to introduce the FROGS database that includes ground-based, satellite and reanalysis products, gridded to a common $1°\times1°$ daily resolution format in support of the above activities. FROGS is first presented by type of observational source. For each precipitation product, we summarize its main characteristics and the regridding methodology used along with some known pros and cons. Section 3 presents some illustrations of the large ensemble of precipitation from FROGS. Section 4 gives the technical details of the database files including how to access the data and how to reference the data using DOI. We end with a conclusion and outlook section.

# 2. The database

## 2.2 The ground-based products

In situ products are seen by many as offering a 'ground truth' but this is not necessarily the case since they are gridded estimates based on various interpolation algorithms which use incomplete station networks of varying quality and density. All products have their pros and cons especially when it comes to extreme precipitation estimates and all are limited in data coverage over certain land regions e.g. Africa. Some include error estimates or quality masks while others do not. Here we have chosen to

include products in FROGS, which offer daily global or quasi-global land estimates. All products are either already available on a 1°x1° grid or have been re-interpolated onto that resolution using simple averaging. The list of products is summarized in Table 1 and below we present an overview of each individual product.

### 2.2.1   NOAA CPC Global Daily analysis

The NOAA CPC Unified gauge-based analysis of global daily precipitation (Chen et al. 2008; Xie et al., 2010) interpolates data from over 30,000 stations over land onto a regular grid with dedicated quality control (QC) and accounting for topographic biases (Xie et al. 2007). The QC is performed through

historical record comparisons and neighbour checks, concurrent radar/satellite observations and utilises numerical model forecasts. The gauge reports come from multiple sources including GTS, COOP and other national and international agencies. The daily analysis is constructed on a 0.125°x0.125° grid over the entire global land area but is released on a 0.5°x0.5° grid for the period since 1979. The high resolution and station density are a key strength of the dataset but the quality of the gauge-based

analysis is poor in data sparse regions and accumulations can differ from country to country based on time of observation leading to potential discontinuities across national boundaries. A possible

inhomogeneity in the early part of the record has also been recorded with a large decrease in total precipitation and number of wet days between 1981 and 1982. There are two versions of the dataset: (i) a "retrospective version" available from 1979-2005 which uses all the available ~30,000 stations and (ii) a "real-time version" which uses ~17000 stations and spans 2006-present. The main changes being over the CONUS region. The two versions remain temporally homogeneous (Xie P-P, Personal communication, 2019). For FROGS, the two 0.5°x0.5° resolution products are merged and are then averaged onto the common 1°x1° grid using a simple arithmetical mean calculation.

### 2.2.2   The Global Precipitation Climatology Centre (GPCC) daily products

The Global Precipitation Climatology Centre (GPCC) builds a suite of gridded precipitation products based on rain gauge measurements and comprehensive quality control (Becker et al., 2013). In particular a full global 1°x1° daily analysis is available as well as a first guess analysis product (Schamm et al., 2014). GPCC products have the advantage that they can access more data than available to other products but due to data restrictions they cannot share raw station data (Becker et al., 2013). Here, the GPCC Full Data Daily V1 (GPCC-FDDv1) product (Schamm et al., 2014) is made available up to December 2013. In GPCC-FDDv1 station data were interpolated using ordinary block Kriging, a stochastic interpolation method which accounts for the statistical structure of precipitation in terms of a distance-weighted spatial autocorrelation function. The daily precipitation estimates represent area averages which results in estimates directly comparable to other forms of data that produce area average estimates such as satellite products. Also included in FROGS is the recent Full Data Daily V2018 (GPCC-V2018) product, which was released in June 2018 and covers 1982 to 2016 (Ziese et al., 2018). In addition to more data and a more advanced quality control, the main difference between GPCC-V2018 compared to GPCC-FDDv1 is that a modified SPHEREMAP interpolation scheme (Becker et al., 2013) is used rather than ordinary block Kriging to align with the method that is applied to GPCC's monthly products. GPCC recommend the GPCC-V2018 product for analyses of extreme events and

related statistics at daily resolution however to date there have been no independent analyses to confirm this. Finally, the first guess product, based on NRT stations is also used for the recent period from 2009-2016.

### 2.2.3   REGEN datasets

REGEN is the name given to a set of daily land-based precipitation datasets created through a collaboration with the University Of New South Wales (UNSW), GPCC and NOAA's National Center for Environmental Information (NCEI). There are two related datasets that are currently available on a 1°x1° daily grid resolution (Contractor et al., 2019): (i) "REGEN-All" which interpolates all available station data and (ii) "REGEN-Long" which only considers stations that have a minimum of 40 years of data (long term stations only). The REGEN products combines the GPCC *in situ* data used to create the 1°x1° GPCC-FDDv1 product described above with data from the Global Historical Climatology Network-Daily (GHCN-D) and other sources which results in unprecedented station density and length of record (since 1950) compared to other existing products. The REGEN-All dataset contains an average of over 50,000 station records per day. The two REGEN products use the ordinary block Kriging algorithm described in Schamm et al. 2014 to interpolate the data onto a 1°x1° grid resolution. The gridded fields are also supplemented with metadata including number of observations, standard deviation, Kriging error and data quality mask. The addition of metadata is a key strength of REGEN-All along with the number of observations utilised. However, the dataset does suffer from a varying station network over time (e.g. stations per day double in North America after 2000; decrease substantially in South America from the late 1990s and in India since the 1970s). However as this is a very new dataset, testing has been limited and so the effect of network changes has not been fully explored. The version V1-2019 of REGEN products is included in the FROGS database.

## 2.3 The satellite-based products

Most of the "satellite" precipitation estimation products make use of ancillary, non-satellite data in their estimations and these enriched products are regarded as the best estimate. Nevertheless, for most of these products, an unadjusted version is also available and is included here. Hardly any dataset is truly global and we present below the quasi-global land and ocean datasets as well as the ocean only, the land

only quasi-global data products currently available in FROGS. The database is completed with some regional products including tropical land and oceans or ones covering only continental Africa and South America. The list of products is summarized in Table 2 as well as information regarding their respective spatial/temporal coverage.

Most of the products are available at the daily scale and so this is the version we use. The day is defined over the 00Z-00Z period for each of the datasets unless otherwise specified. The daily data are then regridded onto a 1°x1° regular grid covering the whole earth. We currently use a simple arithmetical averaging (no interpolation) going from the small scale in degrees to the large scale in degrees that allows a conservative average of precipitation at 1°x1°.

### 2.3.1 The global and quasi-global land and ocean datasets

### 3B42 v7.0 by NASA

The 3B42 v7 product (Huffman et al. 2009; Huffman et al., 2011) is a reference product in various previous studies of tropical rainfall distribution (Maggioni et al., 2016). It also exemplifies what a dataset that is highly geared towards microwave data (imagers and 183GHz sounders) can provide in terms of daily accumulation. It also puts to perspective the use of scattering based retrieval over land for instantaneous retrieval (Gopalan et al., 2010). The combined TRMM radar-imager product (Haddad et

al., 1997) serves as a reference for other microwave instruments prior to merging. Geostationary-based IR imagery is also used in the algorithm to ensure observations when no low earth observing satellites

are available. The technique also relies on a sophisticated bias correction approach that relies on GPCP monthly analysis over land. While this popular product has been evaluated over a very large number of small regions and catchments as well as over the whole tropics for certain metrics (Sun et al., 2017), it still lacks systematic intercomparison with the whole suite of products presented here. As a

consequence, it is included here even though NASA has announced the discontinuation of its production post 2018. As a complement to the gauge-adjusted product, the microwave-calibrated IR estimates and the microwave only estimates are also provided. Along with this reference product, NASA has also been releasing a low latency version called 3B42RT where RT is for real-time for which no gauge data are used. Two versions are provided here, the actual products and the uncalibrated one (Huffman et al.,

2009a). The suite of 3B42 products is at the core of the constellation-based family of satellite rainfall products.

## GsMAP v6 with and without gauges by JAXA

The Global Satellite Mapping of Precipitation product provides high resolution precipitation estimation using satellite observations from multiple platforms (Kubota et al., 2007; Mega et al., 2018). This product is mainly based on microwave estimation of rainfall for a suite of microwave imagers and sounders. The suite of GSMAP products hence belongs to the constellation-based family of satellite rainfall products. The microwave instantaneous rain rate estimates (Aonashi et al. 2009) are propagated

based on Cloud Motion Wind vectors originally derived from IR geostationary imagery to yield to a gridded high resolution precipitation product (Ushio et al. 2009). GsMAP belongs to the morphing based microwave algorithm, like CMORPH (Joyce et al., 2004) or SearchLight (Bellerby, 2013). To complement the satellite-only estimation, the product is further scaled to rain gauges estimates to correct for some bias over land. Two sets of products are included in the database: unadjusted and

adjusted. Two versions of the GsMAP products are provided here: the so-called reanalysis and the near real time versions of the products that differs in the amount of data they used in the processing. Owing

to the production schedule, the homogeneous processing of the reanalysis data has been performed from mid-2000 up to April 2014. As a consequence, we have restricted our use to the processed full years from 2001 to 2013 inclusive. While the original product is offered at two daily average range (00-00Z and 12-12Z), here for the sake of homogeneity, only the 00-00Z daily average is provided in the

ensemble database. The near real time dataset extends up to 2017.

## PERSIANN CDR v1 by NOAA

PERSIANN-CDR v1 is a quasi-global IR-based product, trained over radar data in the US, and normalized to GPCP monthly totals (Ashouri et al., 2015). It can be thought of as an alternative daily

downscaling of the GPCP monthly data to that of GPCP 1DD CDR. Despite sharing monthly totals, the two products differ substantially in their estimation of the daily precipitation distribution (Sun et al., 2017) and so are included in the database. The product does not rely on passive microwave data and as a consequence extends back further in time than GPCP 1DD. PERSIANN-CDR has been extensively evaluated over various regions and was shown to provide mixed levels of agreement with observations

from local rain gauge networks (Miao et al., 2015; Tan and Santo, 2018). PERSIANN-CDR is the climate monitoring oriented product of the PERSIANN family of datasets that can otherwise be accessed at the CHRS data portal (Nguyen et al., 2019).

## CMORPH v1.0 RAW and CRT by NOAA

The CMORPH product (Joyce et al., 2004; Xie et al., 2017) belongs to the microwave based morphing algorithms like GsMAP (Kubota et al., 2007). The microwave derived instantaneous rain rate estimates from multiple platforms are propagated using Cloud Motion Wind vectors originally derived from IR geostationary imagery and a Kalman filter  (Joyce and Xie, 2011). Such an approach results in a high-

resolution precipitation product. The CMORPH products belong to the constellation-based family of satellite rainfall products and both microwave imagers and sounders are used. The instantaneous

imagers based rain rates are obtained using GPROF 2004 (Kummerow et al., 2001) while the sounders

estimation rely on the algorithm of (Ferraro et al., 2005). Two versions are provided with (CRT) or

without (RAW) gauges adjustment. The adjustment is performed using PDF matching. Over land, the

daily CPC gauges analysis is used for this correction (Xie et al., 2003) while over ocean, the adjustment

is done using the GPCP pentad merged product. The product is thought to perform well overall, with a

small bias relative to the gauges; yet experiences difficulty with snow and cold season rainfall (Xie et

al., 2017).

## GPCP 1DD CDR v1.3 by NOAA

The Global Precipitation Climatology Product CDR Version 1.3 daily product (Huffman  et al. 2001) is

another reference product used in various previous studies (Adler et al., 2017). The GPCP CDR dataset

is the only global product in the database. It is adapted from the Geostationary Operational

Environmental Satellite Precipitation Index technique with monthly, local adjustments. The approach

merges IR imagery from geostationary and polar platforms. It relies on the use of one single microwave

platform and the Level 2 retrievals of Kummerow et al. (1996). A bias adjustment scheme is finally

used over land that relies on rain gauge data from the GPCC database at the monthly scale. Over the

high latitudes, GPCP incorporates IR based precipitation estimations in complement to microwave

derived rain rates for the lower latitudes. The original data file from NOAA contains a valid range

attribute between 0-100 mm/d. Values beyond 100mm/d are nevertheless found in the dataset. Two

versions are provided here: 1) one where the valid range attribute is not enforced so original values

extending beyond the valid range are kept in the analysis (Alder, 2019 personal communication) and 2)

a version in which the valid range is enforced.

### 2.3.2 The quasi-global land only dataset

### CHIRP/CHIRPS v2.0 by UCSB

The Climate Hazards Infrared Precipitation (CHIRP) and the Climate Hazards Infrared Precipitation with Stations (CHIRPS) are satellite based precipitation product (Funk et al., 2015). While CHIRP is satellite only, CHIRPS also benefits from station data from 5 public data sources (GHCN monthly and daily, Global Summary Of the Day, GTS daily, Southern African Science Service Centre for Climate Change and Adaptative Land Management) as well as private datasets from various countries in the

world (see Funk et al., 2015 for the list). As a result, the density of gauges in the final product varies significantly in time as well as space. CHIRP uses the infrared observations from geostationary observations in a GOES GPI modified approach and various ground based and alternative sources (an in-house climatology, 3B42, Climate Forecasts Systems outputs) for its calibration at the monthly scale. It is considered as a rain gauges free or satellite only product. Then the CHIRPS estimates are obtained

by merging the stations with the CHIRP estimates using a weighted average of the closest stations and CHIRP results for each 0.05° grid point. The unique characteristics of the CHIRPS product are its native high resolution, low latency and long record (the longest of the satellite data in the database, Table 2). The product has been used in many evaluation and process studies and is thought to support hydrological forecasts and trends analysis in Ethiopia for instance (Pricope et al., 2013). CHIRPS is a

quasi-global implementation of algorithms and methodology that also have been implemented at regional scales like the TAMSAT products (see below).

### SM2RAIN-CCI by ESA

While all of the other products are based on indirect measurements more or less, this product actually relies on very indirect evidence of precipitation by relating satellite-based estimations of soil moisture to the precipitation that affected the surface. This product is based on the SM2RAIN algorithm (Brocca

et al., 2013). The algorithm is applied onto the active and passive ESA Climate Change Initiative soil moisture datasets (Ciabatta et al., 2018). It is an alternative way to use indirect satellite-based measurements to estimate rainfall. Note that due to soil moisture data quality issues, a mask is applied to the rainfall products and no estimates are provided over the tropical rainforest areas, frozen and snow

covered soil, rainforest areas, and areas with topographic complexity.

### 2.3.3 The quasi-global ocean only dataset

### HOAPS v4.0 by CMSAF

The "Hamburg Ocean-Atmosphere Parameters and Fluxes from Satellite Data" product is described at length in (Andersson et al., 2010). This product relies on recalibrated and inter calibrated measurements from SSM/I and SSMIS passive microwave radiometers (Fennig et al., 2017) to estimate a suite of fresh water budget elements globally (80°S-80°N) over sea-ice free ocean surface, including precipitation. Here version 4 of the product is provided (Andersson et al., 2017) but corresponds to version 3.2 for the

precipitation algorithm. The precipitation retrieval is based on a neural network technique that relies on the polarized brightness temperature measurements of the conical scanning imager. The neural network is trained on ECMWF inputs and radiative transfer simulations. Unlike other similar products, HOAPS precipitation appears to detect snowfall well during the cold season (Klepp et al., 2010). The orbit data have been regridded on the common 1°x1° daily grid courtesy of M. Schröder.

### 2.3.3 Tropical land and ocean dataset

### TAPEER v1.5 by AERIS

The recently released TAPEER product is based on the universally adjusted Geostationary Operational Environmental Satellite precipitation index technique (Xu et al., 1999) that merges geostationary

infrared imagery with microwave instantaneous rain rates estimates at daily local scales to yield the daily precipitation accumulation (Kidd et al., 2003). The current implementation relies on the BRAIN L2 dataset (Viltard et al., 2006) for a suite of conical microwave imagers and include the SAPHIR data from the Megha-Tropiques mission (Roca et al., 2015) for rainfall detection and is available at 1°x1°

resolution (Roca et al., 2018). Along with the accumulation, an estimation of the sampling uncertainty of the daily accumulation is provided (Chambon et al., 2012; Roca et al., 2010). The TAPEER product has been favourably compared against various datasets over tropical Africa  (Gosset et al., 2018; Guilloteau et al., 2016). Unlike many other operational satellite precipitation products, the TAPEER estimations do not ingest nor are calibrated to any rain gauge datasets. As such, it provides one solution

independent from the rain gauge network and with an enhanced tropical sampling thanks to the use of the SAPHIR data from the Megha-Tropiques mission. While the original product is offered at four daily average range (00-00Z, 06-06Z, 12-12Z and 18-18Z), here for the sake of homogeneity, only the 00-00Z is provided in the ensemble database. This product belongs to the constellation-based family of satellite precipitation products.

### 2.3.3 Africa land only datasets

### TAMSAT v2.0 and v3.0

The Tropical Applications of Meteorology using SATellite data and ground-based observation (version 2.0 and 3.0 (Maidment et al., 2017), is a product that provides rainfall estimates across Africa based both on geostationary thermal infrared (TIR) images, obtained every 15 min (30 min prior to June 2006), and on ground-based observations from the Global Telecommunications System (GTS). The TAMSAT algorithm is based on two primary data inputs: *i*) Meteosat TIR imagery provided by The

European Organisation for the Exploitation of Meteorological Satellites (EUMETSAT) and *ii*) rain gauge observations (daily accumulated, 0600-0600 UTC) for calibration. The general procedure follows

three steps: a) algorithm calibration - at the decadal (version 2.0) and pentadal (version 3.0) time-step;

b) estimation of the pentadal/decadal rainfall estimates; and c) estimation of daily rainfall estimates. The

TAMSAT daily rainfall estimates have a native resolution of 0.0375° (about 4 km) and covers all of

Africa since January 1983 to the present.

## ARC v2.0

The African Rainfall Climatology, version 2.0 (ARC2) it is a revision of the first version of the ARC

and is consistent with the operational Rainfall Estimation Version 2 (RFE 2.0) (Novella and Thiaw,

2013). The product is a composite of *i*) 3-hourly geostationary infrared (IR) data centred over Africa

from the European Organisation for the Exploitation of Meteorological Satellites (EUMETSAT), and *ii*)

quality-controlled 24-h (0600-0600 UTC) rainfall accumulation records from the Global

Telecommunication System (GTS) gauge database. The calibrated IR and the quality-controlled GTS

gauges are then combined following multiple criteria (i.e., the two-step merging process) to produce the

final rainfall estimates. The ARC2 daily data set is updated regularly. The native resolution is 0.1° x

0.1° over a spatial domain of 40°S–40°N and 20°W–55°E and over the period from 1 January 1983 to

the present.

### 2.3.4 Latin America land only datasets

## COSCH

The Combined Scheme approach CoSch (Vila et al., 2009) is a gauge-satellite-based precipitation

product that provides daily gridded estimates over Latin America. The general procedure for satellite-

gauge merging and data production involves the follow tasks: *i*) obtain and run quality control of global

and regional rain gauge data, from GTS and multiple institutions, respectively; *ii*) reprocess the daily

accumulated satellite-based rainfall fields, following the same time accumulation as the rain gauges (1200-1200 UTC); and *iii*) apply the additive and multiplicative bias correction schemes for each station on a daily basis. The CoSch actual product-version uses the Real Time TRMM Multisatellite Precipitation Analysis (TMPA-RT, Huffman, 2007) (Version 7) as a high-quality satellite rainfall

algorithm. The CoSch daily rainfall estimates database is available from March 2000 to the present and its native spatial resolution is 0.25° over the Latin America land areas.

## 2.4 The reanalysis products

Atmospheric reanalyses blend observed meteorological state fields (temperature, humidity, wind and pressure) with a global weather model through assimilation to provide a continuous representation of not only the state fields, but also the model-generated fields. Precipitation is one such model derived but observationally guided field. Typically, reanalysis precipitation is considered to have more uncertainty than the analysed state fields (Kalnay et al., 1996). However, precipitation is a key quantity in both the

reanalyses representations of global water and energy cycles (through the latent heat of condensation) and so should be understood (Bosilovich et al., 2008). There are few studies inter comparing many reanalyses daily precipitation. Although  distinctly different distributions were found among a collection of 10 analyses and reanalyses (focusing on gauge data over the United States) (Bosilovich et al., 2009). Even for a given weather event, the distribution of the precipitation can have large variance. Shiu et al

(Shiu et al., 2012) results suggest that reanalyses can reproduce the temperature-precipitation relationship as temperature warms, but the more recent reanalyses had higher variance than the older generation. The long-term collection of daily reanalyses precipitation here will help characterize and understand the state of the reanalyses ability to reproduce the high frequency occurrences of extreme precipitation.

The list of products is summarized in Table 3 and below we detail the common grid and present each individual product.

## MERRA and MERRA 2

The Modern-Era Retrospective Analysis for Research and Applications (MERRA) version 1 (Rienecker et al., 2011)  and version 2 (Gelaro et al., 2017) benefited throughout their development from a focus on the water cycle which representation was identified as a key component to understanding weather and climate. Significant improvements were included in the model (Molod et al., 2015) and the water vapor analysis (Takacs et al., 2016). While the influence of observing system changes is still apparent in

MERRA-2 (Bosilovich et al., 2017) (Bosilovich et al. 2017), and there are some significant regional biases (e.g. tropical land topography overestimates), there is indication that the extreme end of the distribution is significantly improved in MERRA-2 over MERRA for the continental United States (Bosilovich and al., 2015). The observations evaluated here will allow the testing of these improvements in other regions in reanalyses.

## JRA 55

The details of the Japanese 55-year Reanalyis (JRA-55) are provided in (KOBAYASHI et al., 2015). This version introduced 4D variational analysis extending in time beyond the introduction of satellite

data for weather analysis (back to 1958). Wind profile retrievals for tropical cyclones were assimilated, and provide a significant contribution to the analysis of tropical cyclones. While some improvements have been noted in the stability of the precipitation time series and certain water vapour biases, the JRA-55 mean precipitation tends to be high, attributed to a dry model bias and spin-down effect of the forecast following reinitialization.

ERA-Interim

The ECMWF Interim Reanalysis (ERA-Interim) (Dee et al., 2011) was developed to test the recent advancement of the forecast model and assimilation development beyond ERA-40 (Uppala et al., 2005), especially in the representation of the hydrologic cycle. This included advances in the humidity analysis, radiance bias correction, and cloud parameterization, which are crucial for the representation of the water vapour state and generation of precipitation. While the large-scale representation of the precipitation has improved over ERA-40, some differences from observed data can be found (Simmons et al., 2010).

CFSR

The National Centers for Environmental Prediction (NCEP) Climate Forecast System Reanalysis (CFSR) was developed to provide initial conditions for continuing seasonal predictions, as well as for climate studies(Saha et al., 2010). At a horizontal resolution of 38km, the representation of the modelled precipitation will be the highest resolution source reanalysis data included here. While high resolution should provide improved locations of precipitation events and structural patterns, the CFSR also uses observation corrected precipitation for forcing its land surface model. This was done to provide the best surface forcing and soil moisture for the subsequent forecasts. As with the other reanalyses here, the influence of changing observations, especially the addition of ATOVS radiances, significantly affects the mean precipitation of CFSR (Zhang et al., 2012).

Regridding method

All of the data for the reanalyses (MERRA, MERRA-2, CFSR, and JRA-55, ERA-Interim) were obtained from the CREATE service (Potter et al., 2018). These data are identically formatted with one variable per file for both 6-hour and monthly. The 6-hour was then used to create the daily form and the data time was adjusted to have a 12-hour mid time. The files were also adjusted to have the same

longitudinal wrap as GPCP. The files were regridded to 1°x1° using a bilinear remapping with the Climate Data Operators (CDO).

## 4  An illustration of the database

Figure 1 shows the annual mean precipitation time series all of the products and indicates the various time spans and spatial coverage of the products. This large ensemble of products is characterized by various trends in their depiction of the average precipitation evolution. Note that the regional products might not be compared directly with the quasi-global ones. Despite this, there are some clear outliers and inhomogeneities in the products available. It is recommended that further work aims at

understanding these differences between the products through a concerted community intercomparison effort.

[Figure 1 here]

## 5  Data Availability

Data format

Files are produced within netCDF-4 format with metadata following the Climate and Forecast (CF) Convention version 1.6 and Attribute Convention for Dataset Discovery (ACDD) version 1.3. An example of the header of a product is provided in the Annex.

One file per product, per year, at the resolution 1° x 1°x 1-day. Each yearly file contains the following information as a minimum:

- **lon**: The longitude values of the grid, in degrees, ranging between [-179.5; +179.5]. The grid is centred, i.e., a value of +0.5 corresponds to the degree [0W, 1W].
- **lat:** The latitude values of the grid, in degrees, ranging between [-89.5; +89.5]. The grid is
25       centred, i.e., a value of +0.5 corresponds to the degree [0N, 1N].
- **time:** The time values of the grid, following the standard CF calendar, in days, centred, since 01-01-1970, midnight.
- **rain:** The precipitation estimate, in [mm/day]. Missing values are represented with NaN_F.

For some products, extra information can be found in the files. For instance, the TAPEER product is completed with an estimate of the uncertainty of the daily precipitation.

Accessibility and DOI

The database (Roca et al., 2019) is referenced with the following DOI:

**http://dx.doi.org/10.14768/06337394-73A9-407C-9997-0E380DAC5598**. The DOI landing page provides the up to date how to access the database as well as a number of useful references for users.

## 3. Conclusions and outlook

For the first time we offer an easily accessible database of daily precipitation products on a common
grid which we hope will prove invaluable for intercomparison, model evaluation (Tapiador et al., 2017, 2018) and other research purposes. In particular, FROGS offers an invaluable resource to study precipitation extremes and to help us understand some of the uncertainties that are inherent across all precipitation products. This understanding should extend to considering resolution and scaling effects on extremes imposed through the gridding of point-based information (e.g. Dunn et al. 2014) and the
regridding to lower resolution of some of the products (e.g. Herold et al. 2017) which could 'smooth' extremes. A few studies based on this database are already under consideration in various journals with a focus on extreme precipitation (Special Issue in ERL). This is a 'living' database, that is, products will continue to be added with time. This includes the latest release of IMERG (Huffman et al., 2015), MSWEP (Beck et al., 2018)  and possibly regional *in situ* products (e.g. APHRODITE, E-OBS, AWAP
etc). How the database continues to evolve will be shaped by the needs of the community and the feedback from various on-going assessments. Similarly, efforts will be geared towards updating the database with the most recent years as they become available.

Acknowledgements

We kindly acknowledge the participants the WCRP meeting on Extreme Precipitation at DWD in Offenbach, 2018 who provided a strong incentive for this database to be finalized. Dr A. Guérou's help at an early stage of this effort is appreciated. All the data providers are also acknowledged for making their data freely available. We further thank the GSMAP, GPCP, GPCC, CPC and CHIRPS teams for enriching exchanges about their products. We thank Drs. M Schroder, A. Ramos for helpful discussions on the data. The graphic has been done using the NCAR Command Language (NCL 2013). MB was supported by Australian Research Council (ARC) Discovery Project DP160103439 and LVA by the ARC Centre of Excellence for Climate Extremes (CE170100022).

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

        float lon(lon) ;

                lon:standard_name = "longitude" ;

                lon:long_name = "longitude;

                lon:units = "degrees_east" ;

                lon:axis = "X" ;

                lon:comment="The longitude values of the grid, in degrees, ranging
between [-179.5; +179.5]. The grid is centred, i.e., a value of +0.5 corresponds to
the degree [0W, 1W]"

        float lat(lat) ;

                lat:standard_name = "latitude" ;

                lat:long_name = "latitude";

                lat:units = "degrees_north" ;

                lat:axis = "Y" ;
```

```
                    lat:comment="The latitude values of the grid, in degrees, ranging
between [-89.5; +89.5]. The grid is centred, i.e., a value of +0.5 corresponds to
the degree [0N, 1N]";

        float rain(time, lat, lon) ;
                    rain:standard_name = "Precipitation » ;
                    rain:long_name = "3B42 v7 Daily Accumulated Rainfall (0h-0h)" ;
                    rain:units = "mm" ;
                    rain:missing_value = NaNf ;
                    rain:_FillValue = NaNf ;
                    rain:comment="TBD";

    // global attributes:
                    :conventions= "CF-1.6, ACDD-1.3" ;
                    :title="TBD";
                    :project=" TBD ";
                    :summary=" TBD "
                    :source="Gridded Daily Accumulated Rainfall derived from TRMM/3B42
    v7 products "
                    :institution="NASA/LEGOS/IPSL"
                    :license="TBD";
                    :aknowledgement="TBD";
                    :creator_name = "NASA" ;
                    :creator_email = "none" ;
                    :creator_url = "https://pmm.nasa.gov/TRMM" ;
                    :creator_type="institution";
```

```
                :creator_institution="NASA";

                :contributor_name = "Rémy Roca" ;

                :contributor_role = "data gridding and formatting in netCDF" ;

                :contributor_email = "none" ;

:contributor_type="person";

                :contributor_institution="LEGOS";

                :publisher_name = "ESPRI" ;

                :publisher_email = "none" ;

                :publisher_type="institution";

:publisher_url = "http://geonetwork..." ;

                :publisher_institution="IPSL"

                :product_version="1";

                :time_coverage_start = "1988-01-01" ;

                :time_coverage_end = "2013-12-31" ;

:time_coverage_resolution = "day" ;

                :geospatial_lat_min = -89.5f ;

                :geospatial_lat_max = 89.5f ;

                :geospatial_lon_min = -179.5f;

                :geospatial_lon_max = 179.5f ;

:geospatial_lat_resolution = 1f ;

                :geospatial_lon_resolution = 1f ;

                :geospatial_lat_units = "degrees_north" ;

                :geospatial_lon_units = "degrees_east" ;

                :cdm_data_type="grid";
```

```
:keywords="GCMD:EARTH SCIENCE,GCMD:ATMOSPHERE,GCMD:PRECIPITATION" ;

:platform="GCMD:Earth Observation Satellites";

:instruments="GCMD:Earth Remote Sensing Instruments,GCMD:Passive
Remote Sensing";

:instrument_vocabulary = "GCMD:GCMD Keywords" ;

:platform_vocabulary = "GCMD:GCMD Keywords" ;

:keywords_vocabulary = "GCMD:GCMD Keywords, CF:NetCDF COARDS Climate
and Forecast Standard Names" ;
```

**Tables**

| Product shortname and version | Period used | Spatial coverage | References |
|---|---|---|---|
| CPC | 1979-2017 | 60°s-90°n | (Xie et al. 2010) |
| GPCC Full Daily v2018 | 1982-2016 | 60°s-90°n | (Schneider et al., 2018) |
| GPCC Full Daily v1 | 1982-2013 | 60°s-90°n | (Becker et al. 2013) |
| GPCC First Guess v1 | 2009-2016 | 60°s-90°n | (Becker et al. 2013) |
| REGEN All V1-2019 | 1950-2016 | 60°s-90°n | (Contractor et al., 2019) |
| REGEN Long V1-2019 | 1950-2016 | 60°s-90°n | (Contractor et al., 2019) |
|  |  |  |  |

**Table 1 The ground-based datasets**

| Product shortname and version | Period used | Spatial coverage | Use of rain gauges data | Use of IR satellite data | Use of MW satellite data rainfall estimate | Main Scientific References and ATBD |
|---|---|---|---|---|---|---|
| **Satellite based quasi global** | | | | | | |
| **3B42 v7.0** | 1998-2016 | 50°s-50°n | Yes | Yes | multiple platforms | (Huffman et al. 2009) |
| **3B42 v7.0 IR** | 1998-2016 | 50°s-50°n | No | Yes | No | (Huffman et al. 2009) |
| **3B42 v7.0 MW** | 1998-2016 | 50°s-50°n | No | No | Yes | (Huffman et al. 2009) |
| **3B42 RT v7.0** | 2000-2017 | 50°s-50°n | Yes | Yes | multiple platforms | (Huffman et al. 2009) |
| **3B42 RT v7.0 uncalibrated** | 2000-2017 | 50°s-50°n | No | Yes | multiple platforms | (Huffman et al. 2009) |
| **GSMAP-RNL-gauges v6.0** | 2001-2013 | 50°s-50°n | Yes | yes | multiple platforms | (Kubota et al., 2007) |
| **GSMAP-RNL-no gauges v6.0** | 2001-2013 | 50°s-50°n | No | yes | multiple platforms | (Kubota et al., 2007) |
| **GSMAP-NRT-gauges v6.0** | 2001-2017 | 50°s-50°n | Yes | yes | multiple platforms | (Kubota et al., 2007) |
| **GSMAP-NRT-no gauges v6.0** | 2001-2017 | 50°s-50°n | No | yes | multiple platforms | (Kubota et al., 2007) |
| **PERSIANN CDR v1 r1** | 1983-2017 | 50°s-50°n | yes | Yes | No | (Ashouri et al., 2015) (Sorooshian et al., 2014) |
| **CMORPH V1.0, RAW** | 1998-2017 | 60°s-60°n | No | Yes | multiple platforms | (Xie et al., 2017) |
| **CMORPH V1.0, CRT** | 1998-2017 | 60°s-60°n | Yes | Yes | multiple platforms | (Xie et al., 2017) |
| **GPCP 1DD CDR v1.3** | 1997-2017 | 90°s-90°n | Yes | Yes | One platform | (Huffman et al. 2001) |
| | | | | | | |
| **Land Only** | | | | | | |
| **CHIRPS v2.0** | 1981-2016 | 50°s-50°n Land only | Yes | Yes | No | (Funk et al. 2015) |
| **CHIRP v2.0** | 1981-2016 | 50°s-50°n Land only | Yes | Yes | No | (Funk et al. 2015) |
| **SM2RAIN-CCI** | 1998-2015 | Global Land only | No | No | No | (Ciabatta et al., 2018) |
| | | | | | | |
| **Ocean only** | | | | | | |
| **HOAPS** | 1996-2014 | ocean only | no | no | multiple platforms | (Andersson et al., 2017) |
| **Satellite based regional** | | | | | | |
| **TAPEER v1.5** | 2012-2016 | 30°s-30°n | No | Yes | multiple platforms | (Roca et al, 2018) |
| **TAMSAT v2** | 1983-2017 | Africa (land only) | Yes | Yes | No | (Maidment et al;. 2017) |
| **TAMSAT v3** | 1983-2017 | Africa (land only) | Yes | Yes | No | (Maidment et al;. 2017) |
| **ARC v2** | 1983-2017 | Africa (land only) | Yes | Yes | No | Novella, N.S. and W.M. Thiaw, 2013 |
| **COSH** | 2000-2018 | 60°s-33°n | Yes | Yes | Yes | Vila et al., 2009 |

**Table 2: the satellite-based datasets**

| Product shortname and version | Period used | References |
|---|---|---|
| **MERRA 1** | 1979-2015 | Rienecker et al. 2011 |
| **MERRA 2** | 1980-2017 | Gelaro et al. 2017 |
| **JRA-55** | 1958-2017 | Kobayahi et al (2015) |
| **ERAinterim** | 1979-2017 | Dee et al. (2011) |
| **CFSR** | 1979-2017 | Saha et al. 2010 |

**Table 3: the reanalysis datasets**

## List of figures

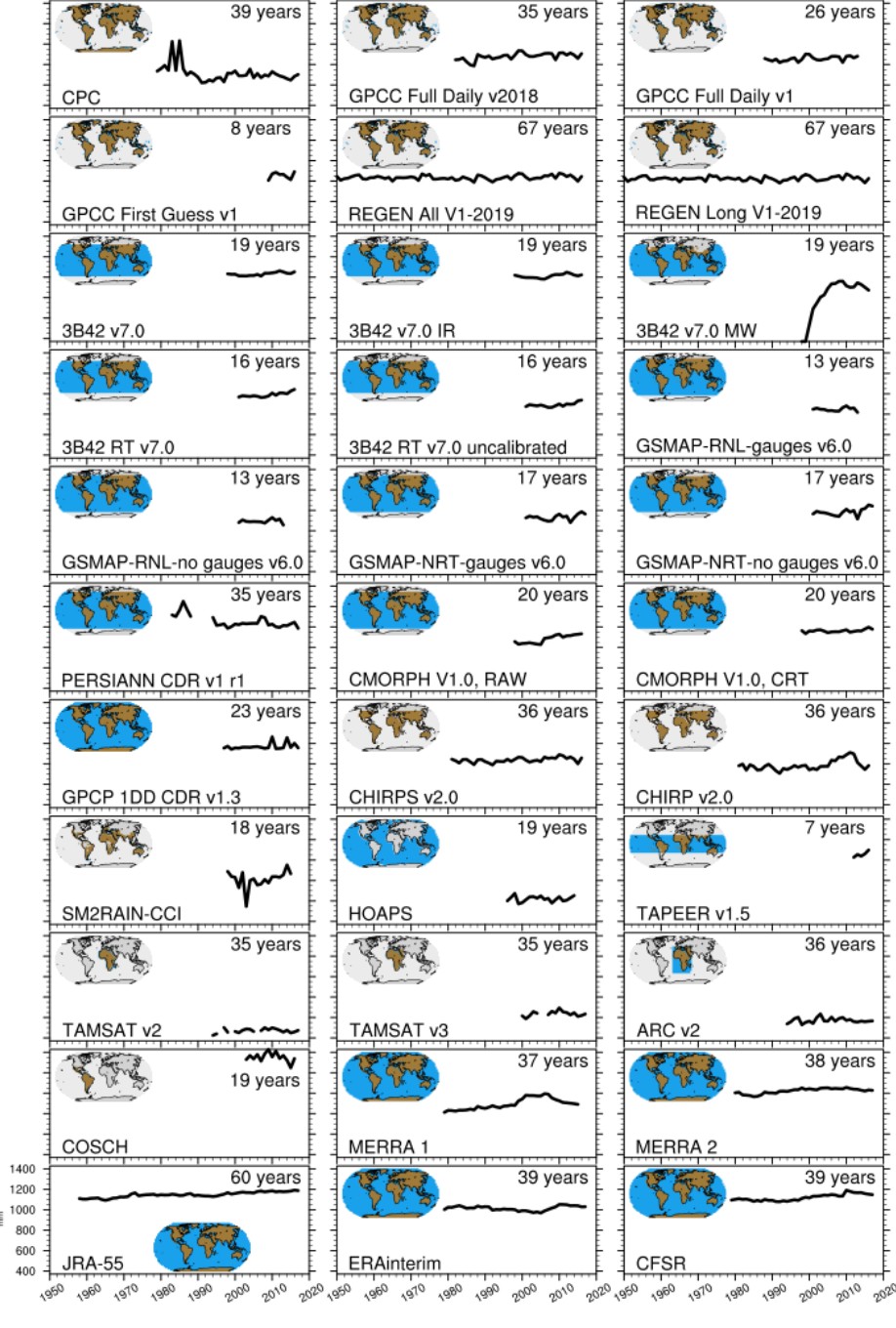

