# Peer review of "FROGS: a daily I°xI° gridded precipitation database of rain gauge, satellite and reanalysis products"

_Earth System Science Data, 2019_

## Referee Comment (RC1) · Anonymous Referee #1 · 2 Apr 2019

This is a very useful paper that describes the FROGs database, a first of its kind. Two issues:

1. The references are a mess. I stopped trying to keep track of what is missing from what is cited. The authors need to vastly improve this.

In addition, there are serious formatting issues with some references.

2. Figure 1 - Can this be improved somewhat? I understand its purpose, yet, there is SO MUCH information in there, it is not very useful. A suggestion would be to show less of the data sets and bound the figure by saying "here is a typical application from FROGs". It would be good to show the legend of the years on all of the charts, as it

stands now, its very meaningless.

---

## Referee Comment (RC2) · Anonymous Referee #2 · 1 May 2019

The purpose of this database is to provide easy access to an ensemble of precipitation datasets (in situ, satellite, reanalysis) that have been re-gridded at a daily 1x1 resolution. The FROGs database (FROGs stands for Frequent Rainfall Observations on GridS). The database includes 6 in-situ datasets, 22 satellite datasets (including 13 global satellite products, 3 land only products, 1 ocean only product, and 5 regional products), and 5 reanalysis products. The authors mention that this database will include additional products in the future. I believe this will be a useful portal. It will allow to access a variety of rainfall products with the same format, spatial and temporal resolutions. This will facilitates global (or local) analysis of precipitation over different period of records.

[Figure]

General comments

1. The fact that the authors include in the database different version of the same product (i.e. GPCC, 3B42, GSMAP, CMORPH, TAMSAT ...) is in my opinion a very good thing. This will allow to compare the products within a same family. The impact of the different retrieval algorithms (i.e. MW vs. IR) or the bias-adjustment procedures (i.e. before/after gauge corrections) could be easily quantified.

2. I have some reservations regarding one of the mentioned use of the database. This application concerns the analyses of extreme events (mentioned on P4, L15-17 and at a couple of other occurrences in the text). The products being re-gridded on the same 1x1 degree grid (i.e. upscaled from their native resolution), there is a possibility that those extreme events will be "washed out" due to the re-gridding procedure (in particular with satellite products going from 0.25x0.25deg to 1x1deg). I wonder if the authors have tested the impact of the re-gridding procedure on precipitation extremes and if they have quantified those differences? In any case, a few words should be added to mention this.

3. At best, the extend of the period of record for the different datasets goes up to the year 2017 (and in one case 2018). Apart for the products that are discontinued, the authors mention the desire to update the database with the most recent year. I think that updating the database at frequent intervals (i.e. once a year) would be useful to the community.

Specific comments

4. P11 and Table 2: Move PERSIANN-CDR up the text to match order in Table 2 (before CMORPH).

5. P17 and Table 3: Move ERA-Interim down the text to match order in Table 3. Add the full product name in the Table.

6. For each Table (1,2,3), I would suggest adding either a column for the native resolution or at least a mark indicating which datasets have been re-gridded at 1x1 deg. resolution (i.e. the datasets that have been modified for the purpose of building this database.

7. Figure 1 is hard to read. This makes sense to try to have all the products in one Figure but I would suggest making each panel bigger (maybe 3 panels by row). The axis label (vertical/horizontal) should be added where needed. Also, it would be better to keep the same order for the products between the text, tables, and within Figure 1 (i.e. the order of the different panels). Additionally, a figure could be added that would include in the same panel a comparison of the datasets belonging to a same category (in-situ, satellite, reanalysis) and same domain (i.e. 50S-50N for instance) (i.e. CPC + GPCC + REGEN, all 3B42 and/or GSMAP, CMORPH, ...). I don't think this would be too much of an effort and this would allow the reader to have a better visual sense of the differences between comparable products (family, domain, type).

8. References: A lot of references (I counted at least 35) are cited in the text but aren't found in the list of references. Vice-versa, a few references cited in the list don't seem to appear in the text.

---

## Author Comment (AC1) · 14 Jun 2019

Dear Editor

We thank the reviewers for their comments and encouragement. Please find below a point-by-point response to the comments. In blue are the comments and in black our responses.

**Reviewer #1**

This is a very useful paper that describes the FROGs database, a first of its kind. Two issues:

1. The references are a mess. I stopped trying to keep track of what is missing from what is cited. The authors need to vastly improve this. In addition, there are serious formatting issues with some references.

This resonates with comment #8 of reviewer #2 too. We apologise for submitting such a reference list. We have now corrected all the references and double checked that all the references mentioned in the text are listed in the bibliography and vice versa. We have also followed a homogeneous style (from ESSD) to format the reference list.

2. Figure 1 - Can this be improved somewhat? I understand its purpose, yet, there is SO MUCH information in there, it is not very useful. A suggestion would be to show less of the data sets and bound the figure by saying "here is a typical application from FROGs". It would be good to show the legend of the years on all of the charts, as it stands now, its very meaningless.

This resonates with comment #7 from reviewer #2. We are reticent however to only show some of the datasets as this might imply that we recommend some over others. Instead we take the suggestion of reviewer #2 to still show all the products but to make the panels bigger and improve the axes in order to make the figure clearer and more meaningful. In addition, we now order the products consistently throughout the text, tables and figures as recommended by reviewer #2.

**Reviewer #2**

The purpose of this database is to provide easy access to an ensemble of precipitation datasets (in situ, satellite, reanalysis) that have been re-gridded at a daily 1x1 resolution. The FROGs database (FROGs stands for Frequent Rainfall Observations on GridS). The database includes 6 in-situ datasets, 22 satellite datasets (including 13 global satellite products, 3 land only products, 1 ocean only product, and 5 regional products), and 5 reanalysis products. The authors mention that this database will include additional products in the future. I believe this will be a useful portal. It will allow to access a variety of rainfall products with the same format, spatial and temporal resolutions. This will facilitates global (or local) analysis of precipitation over different period of records.

1. The fact that the authors include in the database different version of the same product (i.e. GPCC, 3B42, GSMAP, CMORPH, TAMSAT...) is in my opinion a very good thing. This will allow to compare the products within a same family. The impact of the different retrieval algorithms (i.e. MW vs. IR) or the bias-adjustment procedures (i.e. before/after gauge corrections) could be easily quantified.

We thank the reviewer for this statement.

2. I have some reservations regarding one of the mentioned use of the database. This application concerns the analyses of extreme events (mentioned on P4, L15-17 and at a couple of other occurrences in the text). The products being re-gridded on the same 1x1 degree grid (i.e. upscaled from their native resolution), there is a possibility that those extreme events will be "washed out" due to the re-gridding procedure (in particular with satellite products going from 0.25x0.25deg to 1x1deg). I wonder if the authors have tested the impact of the re-gridding procedure on precipitation extremes and if they have quantified those differences? In any case, a few words should be added to mention this.

The reviewer is correct that changing resolution does play a role in the

representation of extremes as outlined for example in Herold et al. 2017, as do other

issues such scale mismatch (i.e. in the case of the gridding in situ data) (Alexander

and Tebaldi, 2012) and the 'order of operation' in which extremes are calculated (e.g.

Avila et al. 2015). All of these issues can in fact smooth the extremes as suggested by

the reviewer. However, as also outlined in Herold et al. 2017, the interproduct spread

of precipitation extremes decreases as the resolution becomes coarser. Therefore we

would expect more agreement between the products at 1degx1deg than at

0.25degx0.25deg when considering precipitation extremes. In fact many of the

precipitation extremes datasets that are used in climate literature for long-term trend analysis, for example, are only available at 2.5degx2.5deg at best (e.g. Alexander et al. 2006, Donat et al. 2013a; 2013b) so having resolutions of 1degx1deg is an improvement on the current state. It's also worth noting that many of the products in FROGs are produced on 1degx1deg already and therefore there is no need to upscale. We expect that researchers would take existing literature into account when they are intercomparing the products and make conclusions about precipitation extremes appropriately. Indeed several papers should appear in a Special Issue in ERL to discuss the appropriateness of the FROGs datasets for extreme precipitation analysis. Based on the reviewers comments, however, we have added the following to the conclusions and outlook section:-

"This understanding should extend to considering resolution and scaling effects on extremes imposed through the gridding of point-based information (e.g. Dunn et al. 2014) and the regridding to lower resolution of some of the products (e.g. Herold et al. 2017) which could 'smooth' extremes."

3. At best, the extend of the period of record for the different datasets goes up to the year 2017 (and in one case 2018). Apart for the products that are discontinued, the authors mention the desire to update the database with the most recent year. I think that updating the database at frequent intervals (i.e. once a year) would be useful to the community.

This is our plan as indicated in the original manuscript line 20 page 20 : "Similarly, efforts will be geared towards updating the database with the most recent years 20 as they become available".

4. P11 and Table 2: Move PERSIANN-CDR up the text to match order in Table 2 (before CMORPH).

Done.

5. P17 and Table 3: Move ERA-Interim down the text to match order in Table 3. Add the full product name in the Table.

We have moved the text as suggested. We have kept the shortname approach in Table 3 (and in the Figure) for consistency with the two other tables. Yet we have spelled out the full names of each reanalysis in the respective paragraphs.

6. For each Table (1,2,3), I would suggest adding either a column for the native resolution or at least a mark indicating which datasets have been re-gridded at 1x1 deg. resolution (i.e. the datasets that have been modified for the purpose of building this database.

We thought about this at the time of writing the manuscript but we have found a few caveats. First, the resolution of the reanalysis is expressed for the spectral model in truncation units but it may actually be different from the available data stream making the definition of the native resolution unclear. Second, as far as the satellite data are concerned we have as much as possible used the already averaged (daily or spatially gridded) products when available so that we started from a resolution of a product that is not the native resolution per se but the resolution of the intermediate product, again making it delicate to state what is the native resolution. Note that in the text we have provided this information for most of the products though. At last, as far as the in situ based gridded datasets are concerned only the CPC archive has been regridded and this is clearly stated in the description paragraph. All in all, we think this detailed information is available in the paper although not in the tables and may not be relevant for reanalysis so we prefer to stick with the actual description of the resolution.

7. Figure 1 is hard to read.  This makes sense to try to have all the products in one Figure but I would suggest making each panel bigger (maybe 3 panels by row).  The axis label (vertical/horizontal) should be added where needed. Also, it would be better to keep the same order for the products between the text, tables, and within Figure 1(i.e. the order of the different panels). Additionally, a figure could be added that would include in the same panel a comparison of the datasets belonging to a same category(in-situ, satellite, reanalysis) and same domain (i.e. 50S-50N for instance) (i.e. CPC +GPCC + REGEN, all 3B42 and/or GSMAP, CMORPH,...).  I don't think this would be too much of an effort and this would allow the reader to have a better visual sense of the differences between comparable products (family, domain, type).

10

As suggested by the reviewer we have made substantial changes to Figure 1 and we hope

15   that it is now easier to read. Changes include reorganizing the panels to make each panel

bigger and follow the exact same order of appearance for the datasets as in the Tables 1,2,3.

We also verified that all products names were similar than the shortnames used in the

Tables.

We thank the reviewer for this suggestion of a second Figure however we believe that it

20   won't necessarily bring as useful information as suggested. Indeed, even if some products of

the same type share a similar spatial coverage (e.g. most of the ground-based datasets) they

might not have the exact same one (e.g. representation of the coastal regions, the islands,

Antartica included or not, etc). Grouping the datasets with regards to a similar spatial

coverage will therefore be very like grouping the different versions of a product together. In

25   this paper we don't want to recommend any version of a product over another one but

rather to present all the products of the database individually. It is worth noting that some

papers submitted in the dedicated Special Issue present such recommendation based on a

more thorough intercomparison.

8. References: A lot of references (I counted at least 35) are cited in the text but aren't found in the list of references. Vice-versa, a few references cited in the list don't seem to appear in the text

See response to comment #1 of reviewer #1